# Parathyroid Hormone-Related Protein/Parathyroid Hormone Receptor 1 Signaling in Cancer and Metastasis

**DOI:** 10.3390/cancers15071982

**Published:** 2023-03-26

**Authors:** Yawei Zhao, Shang Su, Xiaohong Li

**Affiliations:** Department of Cell and Cancer Biology, College of Medicine and Life Sciences, The University of Toledo, Toledo, OH 43614, USA

**Keywords:** PTHrP, PTH1R, PTHrP/PTH1R signaling, cancer metastasis, targeted therapy

## Abstract

**Simple Summary:**

The impact of parathyroid hormone-related protein (PTHrP) and parathyroid hormone receptor 1 (PTH1R or PTHR1) on cancer initiation, growth, and metastasis has been extensively documented in a number of in vitro and in vivo studies. Despite these findings, the attempts to target PTHrP/PTH1R signaling in cancer therapy have not produced successful results in the clinical setting. In light of these conflicting data and conclusions, this review seeks to provide a comprehensive examination of the role of PTHrP/PTH1R in cancer progression and metastasis, as well as offer insights for future research efforts in this field.

**Abstract:**

PTHrP exerts its effects by binding to its receptor, PTH1R, a G protein-coupled receptor (GPCR), activating the downstream cAMP signaling pathway. As an autocrine, paracrine, or intracrine factor, PTHrP has been found to stimulate cancer cell proliferation, inhibit apoptosis, and promote tumor-induced osteolysis of bone. Despite these findings, attempts to develop PTHrP and PTH1R as drug targets have not produced successful results in the clinic. Nevertheless, the efficacy of blocking PTHrP and PTH1R has been shown in various types of cancer, suggesting its potential for therapeutic applications. In light of these conflicting data, we conducted a comprehensive review of the studies of PTHrP/PTH1R in cancer progression and metastasis and highlighted the strengths and limitations of targeting PTHrP or PTH1R in cancer therapy. This review also offers our perspectives for future research in this field.

## 1. Introduction

PTHrP (parathyroid hormone-related protein) is a protein initially isolated from tumors of patients with humoral hypercalcemia of malignancy (HHM) [1]. Although undetectable in healthy human blood, PTHrP is expressed in diverse normal cells and tissues, including the cardiovascular–renal system, lungs, bladder, uterus, placenta, mammary glands, stomach, pancreas, bone, cartilage, and teeth [2]. This suggests that PTHrP may have far more widespread physiological importance than previously thought. In mammary gland development, PTHrP is secreted by the epithelial cells in the embryonic mammary buds, and is necessary for the normal proliferation and differentiation of the surrounding mesenchymal cells of the mammary buds [3]. Loss of PTHrP can cause developmental defects in mammary mesenchymal cells [4]. In cartilage and bone development, PTHrP is expressed in chondrocytes and osteoblasts, and is indispensable for chondrocyte proliferation, osteoblast formation, and subsequent bone formation [5]. In mice, PTHrP deficiency leads to defective rib cage formation and death after birth [6]. Newborn PTHrP haploinsufficient mice showed low bone mass due to decreased bone formation and increased osteoblast apoptosis [7].

The association of PTHrP with the syndrome of HHM initially drew attention in the early 1990s [8]. However, since then, controversial results have been observed and reported in both basic [9,10] and clinical studies [11,12]. In mouse models of breast cancer [13,14] and lung cancer [15], monoclonal neutralizing antibodies (murine [16] or humanized [17]) against PTHrP (1–34) were found to inhibit bone metastasis. Therefore, two clinical trials (NCT00051779 and NCT00060138) were initiated in 2003 to evaluate the safety, tolerability, and possible effectiveness of the humanized monoclonal antibody against PTHrP in patients with breast cancer metastatic to bone, compared to zoledronic acid. However, no results have been reported since the trials were completed in 2004. Furthermore, studies in vitro and in vivo, genetically and pharmacologically, have revealed PTHrP’s causative effects in cancer progression of breast cancer [18], giant cell tumor of bone [19], prostate cancer [20], pancreatic cancer [21], and kidney cancer [22,23]. These findings suggest that PTHrP could be a potential anticancer drug target, and the most effective approach for blocking PTHrP is through monoclonal neutralizing antibodies.

The PTHrP protein exerts downstream effects primarily through binding with PTH1R, a family B GPCR, and the only receptor for PTHrP. Upon binding, PTH1R transfers the signal from the ligands and activates the downstream cAMP signaling pathways. While PTH1R mediates the major actions of PTHrP, the role of PTH1R in cancer and the effectiveness of blocking it have been studied less, compared to PTHrP, partially because of its significant role in bone. PTH and PTHrP can have anabolic or metabolic effects on bone, depending on the doses and times of treatment, through the activation of PTH1R [24,25,26,27,28,29]. In 2002, intermittent treatment of recombinant PTH (1–34) (teriparatide) was approved by the FDA as the first anabolic drug for osteoporosis [30,31,32,33,34]. However, class B GPCRs, including PTH1R, are notoriously difficult to target, especially by small molecule drugs. The structure and dynamics of the active human PTH1R were not resolved until 2019 [35,36]. Nevertheless, there is considerable interest in developing basic knowledge and achieving the translational potential for this class of GPCRs [37,38,39,40,41]. In general, GPCRs comprise 35% of the current clinical drug targets [42]. Peptide agonists and antagonists are relatively advanced in the development pipeline [43,44,45,46,47], with one PTH1R antagonist, (Asn^10^,Leu^11^,D-Trp^12^)-PTHrP (7–34) amide, having been shown to be effective in clear cell renal carcinoma [22,23]. In addition to its role in bone, PTH1R has also been shown to mediate cachexia, with adipocyte-specific *Pth1r* knockout conferring resistance to cachexia driven by kidney failure and lung cancer [48]. Conversely, intermittent treatment with recombinant PTH (1–34) (i.e., teriparatide) decreased bone metastasis of breast cancer, and prolonged survival in a mouse model [49]. 

To gain a better understanding of the role and mechanism of PTHrP/PTH1R in pathophysiology, particularly in cancer, a comprehensive review of the current knowledge on key genes and proteins, signaling pathways, regulation, and mechanisms of action has been conducted. By assessing the strengths and limitations of these studies, we aim to shed light on the potential for targeting PTHrP/PTH1R in cancer for better efficacy and eventual translation into clinical practice. This review aims to contribute to the current knowledge of PTHrP/PTH1R in cancer by summarizing the existing literature on the topic and identifying knowledge gaps and future directions for research. The ultimate goal is to advance the development of effective therapeutic strategies that can improve patient outcomes in cancer treatment.

## 2. The PTHrP and PTH1R Proteins 

### 2.1. The Isoforms of PTHrP

The PTHrP protein has three isoforms, namely, 1–139, 1–141, and 1–173 in length, which result from transcript variants due to alternative splicing [50]. Each isoform is composed of the N-terminal, mid-region, and C-terminal domains (Figure 1). Post-translational processing at multibasic endoproteolytic sites also generates the N-terminal (residues 1–34), mid-region (residues 66–94, 88–106), and C-terminal (residues 107–139, 107–111, 122–139) mature secretory forms of PTHrP [51]. The first 36 residues of the N-terminal fragments of PTHrP in the corresponding full-length, mature polypeptide chain are functional determinants for the interaction with its receptor, exhibiting PTH-like properties in bone, kidney, and cardiovascular systems [52]. Interestingly, both the mid-region and C-terminal domains have independent biological roles. The mid-region PTHrP, i.e., amino acids (AAs) 38–106, contains a nuclear localization signal (NLS) at AAs 66–106. The NLS mediates nucleus and nucleolar translocation for both the full-length and the mid-region PTHrP to act in an intracrine manner [53,54]. The PTHrP in the cytoplasm can translocate to the nucleus by forming complexes with importin, while the secreted PTHrP can translocate to the cytoplasm through endocytosis [55]. These nuclear proteins have been associated with activating the cell cycle, inducing the proliferation of vascular smooth muscle cells, and prolonging the survival of chondrocytes under apoptotic stimulation [54,56]. However, the mechanisms underlying these effects of PTHrP in the nucleus remain to be defined. Interestingly, PTHrP was reported to bind with RNA directly through NLS in the nuclei, predicting its role in regulating RNA metabolism, such as inhibiting rRNA synthesis [57]. This suggests that PTHrP has a far-reaching role beyond being a hormone and a ligand. The fragments corresponding to the C-terminal portion of PTHrP were shown to regulate bone resorption [58] and β-arrestin binding [59]. Engaging both G proteins and arrestins is not solely the function of the C-terminal portion of PTH1R. The PTH1R core, i.e., the external linkers (ELs) between transmembrane helices that opens upon receptor activation, was also shown to have the same function, especially after binding with EL modifying proteins [60].

### 2.2. The Protein Isoform and Downstream Signaling of PTH1R

The PTH1R receptor is composed of three functional domains. The N-terminal extracellular domain (ECD) is responsible for recognizing and binding to ligands. The middle domain of seven transmembrane helices transmits the hormone-binding signal and interacts with the G protein. The C-terminal intracellular domain receives the signal and subsequently interacts with G proteins to activate downstream signaling pathways (Figure 2) [62]. The receptor has a relatively large amino ECD, which plays a crucial role in initial ligand binding, while the seven helical transmembrane domains and connecting loops mediate agonist-induced receptor activation and signal transduction events. The C-terminal tail contains sites involved in mediating ligand-induced receptor internalization, trafficking, and signal termination events. In response to PTHrP, PTH1R can trigger diverse downstream signal transduction pathways, including Gαs/cAMP/PKA, Gαq/PLC/PKC, the Gα12/13/RhoA/PLD pathway, and ERK-1/2/MAPK (Figure 2). The coupling of PTHrP/PTH1R with ERK/MAPK can go through either the G protein-dependent or the G protein-independent/β-arrestin-dependent pathway [63,64]. The structure of the complex of PTH1R with β-arrestin1 was recently deduced from cross-linking two proteins bearing unnatural amino acids in the environment of the living cells [65]. Note that other GPCRs, such as D prostanoid receptor-2 (DP2, Gi-coupled), orphan GPR17 (Gi/q-coupled), and free fatty acid receptor-2 (FFA2, Gi/q/12-coupled), β-arrestin cannot facilitate ERK activation in the absence of functional G proteins [66].

Both ligand structure–activity relationship and receptor mutagenesis studies have revealed that the N-terminal (1–34) of PTHrP interacts with PTH1R via a two-component mechanism, as illustrated in Figure 3 [67]. The segment located at approximate AAs 12–34 interacts with the N-terminal extracellular domain of PTH1R, which is referred to as site one. On the other hand, the segment at AAs 1–12 interacts with the transmembrane helices and extracellular connecting loops of PTH1R, and is referred to as site two. The interaction of PTHrP 12–34 and the ECD of PTH1R accounts for the majority of the binding affinity, while the interaction between PTHrP 1–12 and site two is responsible for inducing the conformational change of the receptor, which in turn initiates the downstream signaling pathway [68,69].

GPCR signaling and function have long been believed to be exclusively at the cell surface. Since the breakthrough discoveries of nuclear binding sites for their ligands in 1980s, many GPCRs, including PTH1R, have been detected in the cell nuclei [70,71,72,73,74]. Studies have shown that PTH1R was present in the nuclear/nucleolar compartment. Positive nuclear staining of PTH1R was shown in various rat tissues, such as the kidney, liver, small intestine, uterus, and ovary, using immunohistochemistry with the antibody validated in PTH1R knockout mice [75]. Moreover, the cellular localization of PTH1R in MC3T3-E1 cells varies during different phases of the cell cycle. It is predominantly present in the nucleus at the early interphase, G0/G1, S, and G2 phases, but it decreases to an undetectable level at prophase and metaphase. The nuclear accumulation remains low until the late phases of telophase, and the expression pattern becomes similar to that during interphase. Serum starvation induces PTH1R nuclear localization, and adding serum back to starved cells restores PTH1R to the cytoplasm [76]. Cytoplasmic localization can also be induced by PTHrP treatment. The active nuclear transport, cell cycle-dependent localization, and response to stimuli suggest that the nuclear localization of PTH1R is a critical factor for cellular function. Therefore, more research is needed to elucidate the mechanism of PTHrP/PTH1R beyond their well-known interaction at the cell membrane. 

Yet, to date, mechanisms of the nuclear translocation remain poorly understood. Nevertheless, the subcellular trafficking of GPCRs is regulated by members of the Ras superfamily of small GTPases [74]. The nuclear localization of PTH1R is possibly facilitated by a conserved potential NLS at the C-terminus 471–487, because the conservation of the peptide sequence was observed across species. Mutation of the C-terminal tail or removal of the residues 475–494 resulted in 50–60% reduction in the internalization of the ligand-bound receptor [77]. In addition, a putative bipartite NLS has been predicted for PTH1R [78], which suggests that importin α1 and β may be associated with nuclear translocations [79]. Conversely, PTH1R was found to coimmunoprecipitate with chromosomal region maintenance 1 (CRM1), indicating a possible mechanism for mediating the nuclear export of PTH1R. Inhibition of CRM1 caused the accumulation of PTH1R in the nucleus [76].

### 2.3. The Conformational Changes of PTH1R during Activation and Recycling

PTH1R is the primary receptor for both PTHrP and PTH, while PTH2R is also capable of binding with PTH to activate the downstream signaling. PTH1R activates several signaling pathways, including the Gα_S_–adenylyl cyclase–cAMP–protein kinase A (PKA), the Gα_q_–phospholipase C (PLC) β–inositol triphosphate–cytoplasmic Ca^2+^–protein kinase C pathway, the Gα_12/13_– phospholipase D (PLD)–transforming protein RhoA pathway, and the β-arrestin–extracellular signal-regulated kinase 1/2 (ERK1/2) pathway [63,80] (see Figure 2). The distinct structures of PTH and PTHrP allow for preferential binding to two different receptor conformations, R^0^ and R^G^, respectively. R^G^-selective ligands, such as PTHrP (1–36), induce transient cAMP responses derived from signaling complexes localized at the plasma membrane, whereas R^0^-selective ligands, such as PTH (1–34) and M-PTH (1–14) (M = Ala/Aib^1^,Aib^3^,Gln^10^,Har^11^,Ala^12^,Trp^14^, or Arg^19^), can induce prolonged cAMP responses that are derived from complexes associated within endosomes [43]. The preferences of PTH and PTHrP for binding to the PTH1R conformations are still unclear, but studies have explored R^0^ and R^G^ selectivity by developing PTH- or PTHrP-specific analogs [81,82]. 

The transient cAMP response is due to the desensitization of the receptor through β-arrestin binding and internalizing the activated PTH1R to the plasma membrane to recycle, or traffic to lysosomes for degradation. On the other hand, stabilization of the β-arrestin–PTH/PTH1R complex can prolong the generation of cAMP and activating ERK1/2 when still in the endosome. The dissociation of β-arrestin from the PTH/PTH1R complex terminates cAMP production, and the PTH/PTH1R-retromer complex is formed after the dissociation of β-arrestin [83]. The retromer complex is involved in late-stage endosomal sorting and retrograde trafficking of vesicles through the Golgi to the plasma membrane. A negative feedback loop involving cAMP, PKA, and the vacuolar ATPase (vATPase) causes the exchange of β-arrestin for the retromer complex [84]. Specifically, intracellular cAMP activates PKA, which phosphorylates and activates vATPase proton pumps. These proton pumps progressively acidify the endosome when it moves along the endocytic pathway, leading to the dissociation of PTH–PTH1R complexes and the termination of the signaling [82,84].

## 3. PTHrP/PTH1R Signaling in Tumor Progression and Metastasis

### 3.1. Diverse Mechanisms of Action Revealed from Preclinical Studies 

In vitro studies have shown that synthetic PTHrP (1–34) peptide or overexpression of full-length *Pth1r* significantly induces cancer cell proliferation and promotes survival in various types of cancer, including prostate cancer [48,49,50], renal carcinoma [78], and breast cancer [85]. The mechanisms of action, including autocrine, intracrine, and paracrine, have been demonstrated with or without binding and activation of PTH1R. In vivo studies have shown that PTHrP plays a causative role in cancer progression, including breast cancer [18], giant cell tumor of bone [19], prostate cancer [20], pancreatic cancer [21], and kidney cancer [22,29]. In a mouse model, overexpression of PTHrP (1–87) or PTHrP (1–173) in the DU-145 prostate cancer cell line significantly increased bone metastasis and caused more severe osteolytic/osteoblastic-mixed lesions in an intrafemoral injection. Interestingly, more bone lesions but lower serum PTHrP were observed in mice injected with PTHrP (1–173) overexpressed DU-145 cells compared to those injected with PTHrP (1–87) overexpressed cells, highlighting the functional pleiotropism of the different PTHrP domains [86]. The independent roles of the NLS and the C-terminus of PTHrP in cancer cells, if any, remain to be determined.

PTHrP mRNA is also detected in human and rat osteogenic sarcoma cell lines, suggesting that PTHrP has an autocrine function in osteosarcoma (O.S.) [81,87,88,89]. Recent studies using mice, in which O.S. was generated by osteoblast-specific deletion of *p53* and *Rb*, showed that both primary tumors and metastasis expressed functional PTH1R [90,91,92]. Genetic knockdown of *Pth1r* in primary O.S. cells reduced proliferation, invasion, and the expression of RANKL in vitro and profoundly inhibited O.S. tumor growth, but increased differentiation/mineralization of the O.S. tumor cells in vivo [90]. In normal osteoblasts, *Pth1h* knockdown or *p53*-deficiency did not affect cell proliferation. However, ablation of PTHrP or CREB induced O.S. cells growth arrest and apoptosis, suggesting that O.S. depends on continuous activation of the PTHrP/PTH1R/PKA pathway [93,94]. This conclusion was further supported using transgenic mouse models, sequestering p53 and Rb, resulting in persistent activation of PKA and causing O.S. initiation and progression [82,95]. 

Moreover, PTHrP/PTH1R was found to mediate drug resistance in prostate cancer bone metastasis, possibly through facilitating TGFβ type II receptor (TGFBR2) degradation [96]. Blocking PTH1R rescued TGFBR2 protein levels in osteoblasts and overcame enzalutamide resistance in a coculture system of prostate cancer and osteoblast cells, suggesting PTH1R as a novel target to overcome enzalutamide resistance in prostate cancer bone metastasis. The remaining questions include whether PTHrP/PTH1R plays roles in other drug resistance, the translational potentials of blocking the signaling, which is the better target to block, and what is the better drug among neutralizing antibodies, peptide antagonists, or small molecule inhibitors.

### 3.2. Opposite Effects in Tumor-Associated Angiogenesis

Over the past two decades, research has investigated the effects of PTHrP on tumor-induced angiogenesis. Early reports showed that PTHrP inhibits endothelial cell migration and angiogenesis by activating PKA, and inhibiting PKA can reverse the antimigratory and antiangiogenic effects of PTHrP [97]. Similarly, PTHrP (1–34), which lacks the NLS and acts by binding to PTH1R, was found to inhibit VEGF expression during endochondral bone formation and osteoblast differentiation [98]. However, there have also been reports of the stimulating effects of PTHrP in tumor-induced angiogenesis. For example, PTHrP can induce the expression of proangiogenic factors, such as VEGF, in breast cancer bone metastasis through PKC-dependent activation of an ERK1/2 and p38 signaling pathway [99]. Additionally, overexpression of PTHrP in pituitary tumor cells can induce neovascularization of the xenografts [100]. Mechanistically, recombinant PTHrP (1–34) increases the capillary formation of endothelial cells through PTH1R activation and cAMP signaling [98]. Possible explanations for these contradictory findings include different cells, such as normal vs. cancer-associated endothelial cells, that responded to PTHrP within the tumor microenvironment. 

The pleiotropic actions of PTHrP in endothelial cells and stem-like cells for angiogenesis may be due to its different domains and biologically active fragments, but more research is needed to explore this possibility. While most studies have shown that PTHrP promotes cancer cell proliferation, tumor progression, and metastasis, one study found that intermittent PTH (1–34) inhibited breast cancer bone metastasis, consistent with the dose- and time-dependent anabolic and metabolic effects of PTHrP in bone [49]. Therefore, caution, precision, and careful data interpretation are needed to draw study conclusions for future translation in patient care. Furthermore, more studies are necessary to uncover the PTHrP domain-specific effects and the respective autocrine, intracrine, paracrine, and endocrine signaling.

### 3.3. Consensus Has Yet to Be Reached: The Elusive Readouts of Targeting PTHrP/PTH1R in Clinical Studies

Breast cancer clinical studies have generally supported the stimulatory effects of PTHrP in tumor growth and progression [13,14,18]. Two genome-wide association studies have also found the *PTHrP* gene, *PTHLH*, in a susceptibility locus for both ER^+^ and ER^-^ breast cancer [101]. In prostate cancer, PTHrP expression was detected in 33% of benign prostate hyperplasia, 87% of well-differentiated tumors, and 100% of poorly differentiated and metastatic tumors, including bone metastatic tumors [102]. PTH1R was found to correlate with reduced overall survival in breast cancer patients [98] and was detected in 37% of primary tumors, but 81% of the bone metastasis samples, supporting the role of the PTHrP/PTH1R system in breast cancer bone metastasis. Another study found coexpression of PTHrP and PTH1R in primary and metastatic cancer cells in matched primary and bone metastatic tissue from patients with untreated adenocarcinoma of the prostate [103]. The high frequency of PTHrP/PTH1R activation in metastasis suggests the significant role of autocrine PTHrP/PTH1R in metastasis. In early-stage lung adenocarcinoma, a positive correlation was observed between PTHrP (1–34) expression and worse overall survival, independent of tumor stage, and coexpressing high levels of N-terminal PTHrP and PTH1R dramatically reduced patients’ overall survival [104].

However, a prospective study over ten years of 402 breast cancer patients found that PTHrP was expressed in 79% of the primary tumors and was positively associated with decreased bone metastasis and improved survival [12]. Nevertheless, bone metastasis still occurred in patients with PTHrP-positive primary tumors, and interestingly, patients with PTHrP-negative primary tumors also developed PTHrP-positive bone metastasis [12]. These findings suggest no direct correlation between PTHrP expression in primary tumors and bone metastasis. Another breast cancer study consistently revealed that PTHrP levels were lower in malignant tissues than in normal breasts, and low nuclear-localized PTHrP correlated with unfavorable clinical outcomes. Nuclear PTHrP levels also correlate positively with nuclear pStat5 levels [102]. Since Stat5 expression and activation in breast cancer correlated with poor prognosis, it was expected that nuclear PTHrP would be associated with worse clinical outcomes [105,106]. However, further studies are needed to determine how these two factors regulate each other, either directly or indirectly. 

Studies have demonstrated that in the early stage of tumorigenesis, PTHrP expression was associated with better survival rates and decreased metastasis in patients with various types of cancer [12,107,108,109], suggesting a tumor-suppressive role of cell-autonomous actions of PTHrP. In later stages, when cancer cells metastasized to the bone, increased PTHrP by the colonized tumor cells mainly stimulated the secretion of RANKL by osteoblasts to induce the osteoclast-mediated bone resorption, which releases factors from the bone matrix to promote the growth of metastasized tumors further [110,111]. These studies showed the tumor-promoting role of paracrine actions of PTHrP and explained the clinical findings, indicating the correlation between PTHrP, reduced overall survival, and decreased metastasis [112,113]. However, the controversial conclusions of these clinical studies on PTHrP warrant further development of experimental tools, such as specific antibodies, patient stratification, larger patient pools from multiple centers, and better experimental design and data interpretation. 

### 3.4. Potential Targeting PTHrP/PTH1R Indirectly in Tumor Progression and Metastasis

In addition to directly inhibiting PTHrP or PTH1R, other approaches have been explored in cancers, including targeting upstream regulators of the PTHrP expression. For example, Wnt signaling drives PTHrP expression in highly osteolytic cancer cells, making it a potential therapy for preventing tumor-induced bone destruction and metastatic outgrowth [114]. However, inhibiting the Wnt pathway in cancer and metastasis is a complex challenge that goes beyond PTHrP/PTH1R signaling.

In breast cancer, TGF-β can induce PTHrP secretion by upregulating Gli2 [115,116]. Inhibition of Gli2 significantly reduces MDA-MB-231 cells-induced osteolytic bone lesions [115]. In clinical trials (NCT00833417, NCT01108094), inhibition of TGF-β and Gli has been evaluated [117,118]. Similarly, EGF can induce PTHrP production, and treatment with EGF receptor tyrosine kinase inhibitors, erlotinib or gefitinib, significantly decreased PTHrP expression and tumor-induced osteolysis in non-small cell lung cancer cells [119,120]. However, these upstream regulators have broad and significant roles in cancer and metastasis beyond PTHrP/PTH1R signaling, raising concerns about the specificity of the inhibitions.

In contrast, one recent study demonstrated the potential of the downstream effectors of PTHrP. Overexpression of PTHrP (1–139) in breast cancer MCF-7 cells suppressed the expression and downstream signaling of LIFR, a reported breast cancer tumor suppressor and dormancy marker [121]. Consequently, the downregulated LIFR promoted MCF-7 cell exit from dormancy, suggesting that inducing and maintaining LIFR, a downstream factor of PTHrP/PTH1R, might keep tumor cells in a dormant state to prevent overt bone metastasis in breast cancer. 

Overall, more research is needed to understand the context-dependent downstream signaling pathways and identify novel targets for effective therapies.

## 4. Conclusions

The roles of PTHrP/PTH1R signaling in cancer progression, metastasis, and tumor dormancy are context-dependent. Both PTHrP and PTH1R have multiple biologically active domains, and their isoforms can localize in the cytoplasm or the nucleus, leading to diversified roles in different cell types. Ligand binding and activation of PTH1R induce various downstream signaling pathways, including cAMP, PKA, PKC, and CREB, which can have different or opposite effects. Moreover, distinct signaling effects can be induced by effectors that control the PTHrP/PTH1R protein degradation and recycling. The response of PTHrP/PTH1R target cells, including cancer cells and different cells in the tumor microenvironments, can differ depending on the stage of disease progression.

Neutralizing antibodies, peptide antagonists, and small molecule drugs have been developed against distinct domains or motifs of PTHrP/PTH1R complexes. However, most of these compounds have limited binding affinity and oral stability, resulting in suboptimal efficacy. Nevertheless, these studies provide good starting points for the optimization of novel binding reagents [122]. Recently released cryo-EM structures of PTH-PTH1R-Gs and PTHrP-PTH1R-Gs complexes further contributed to the better development of novel binding reagents [123,124]. We anticipate the development of selective PTHrP/PTH1R modulators that adopt the advantages of emerging chemical modalities to overcome the limitations of current ones. For example, structural modifications including incorporation of unnatural amino acids (i.e., D-AA), retro-inverso isomers, N-methylation, or stapling reactions between amino acids may result in better PTH1R binding molecules with comparable binding specificity/selectivity and in vivo stability [125]; PTH1R inhibitors can be conjugated with molecules targeting E3 ligase or lysosome to induce PTH1R degradation via proteasome or lysosome; a DNA aptamer interrupting PTHrP binding, but not affecting PTH binding to PTH1R, can be screened de novo based on the cryo-EM structures, and the aptamers can be further optimized for in vitro and in vivo safe delivery. Our prospective targeting approaches against PTHrP/PTH1R are depicted in Figure 4. We expect that successful development of these novel reagents will significantly advance the understanding of the context-dependent roles of PTHrP/PTH1R, and eventually translate to the clinic to target PTHrP/PTH1R in cancer and metastasis. Of course, challenges always accompany opportunities, such as concerns regarding the permeability of the conjugate of PTH1R inhibitor and E3 ligase targeting molecule, the efficacy of this approach in degradation GPCRs, etc. However, several cases have been reported in the degradation of other membrane proteins [126]. Furthermore, membrane-bound E3 ligases may be exploited in the near future [127]. We expect that successful developments of these novel reagents will significantly advance the understanding of the context-dependent roles of PTHrP/PTH1R, and eventually translate to the clinic to target PTHrP/PTH1R in cancer and metastasis.

## Figures and Tables

**Figure 1 cancers-15-01982-f001:**
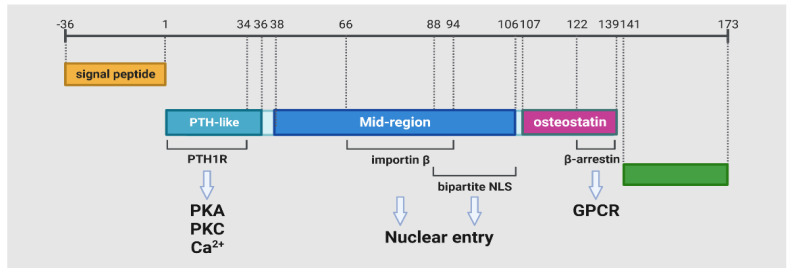
The functional domains of the PTHrP protein. The signal peptide module serves to dock the nascent peptide to the secretory pathway. “PTH-like” indicates the receptor-binding region, and “Mid-region” contains the bipartite NLS essential for PTHrP nuclear import; the C-terminal domains include “osteostatin”, which is involved in bone turnover and β-arrestin binding. The green region is part of the C-terminal that only exists in PTHrP 1–173 [61]. PTH: parathyroid hormone; PTHrP: parathyroid hormone-related protein; PKA: protein kinase A; PKC: protein kinase C; NLS: nuclear localization signals; GPCR: G-protein-coupled receptors.

**Figure 2 cancers-15-01982-f002:**
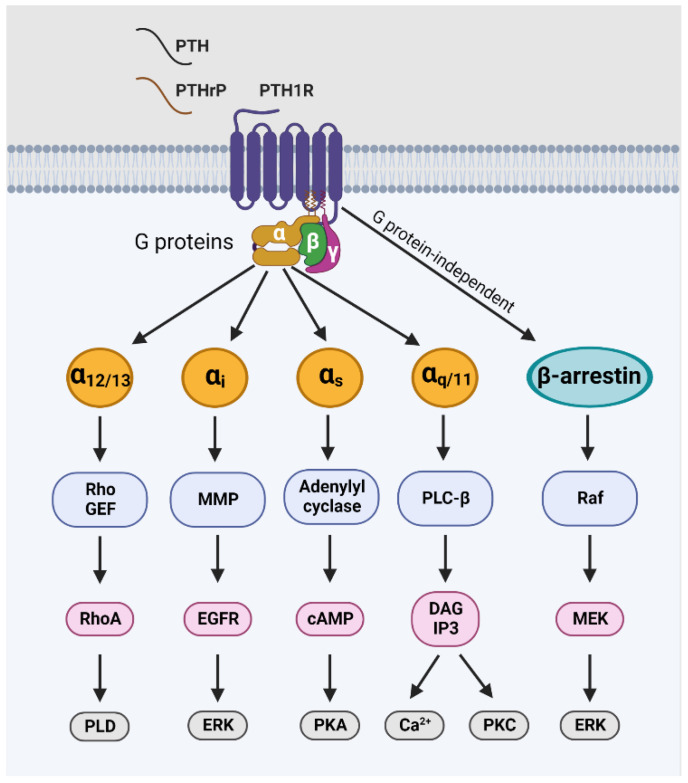
Diagram of ligand binding with PTH1R and the downstream signaling pathway. When PTH or PTHrP binds to PTH1R, it induces a conformational change in the receptor, leading to the activation of diverse downstream signal transduction pathways, via either G protein-dependent or -independent/β-arrestin-dependent mechanisms [52]. GEF: guanine nucleotide exchange factor; PLD: phospholipase D; MMP: matrix metalloproteinase; EGFR: epidermal growth factor receptor; MEK: mitogen-activated protein kinase; ERK: extracellular signal-regulated kinase; PKA: protein kinase A; PKC: protein kinase C; PLC-β: phospholipase C-β; DAG: diacylglycerol; IP3: inositol 1,4,5-triphosphate.

**Figure 3 cancers-15-01982-f003:**
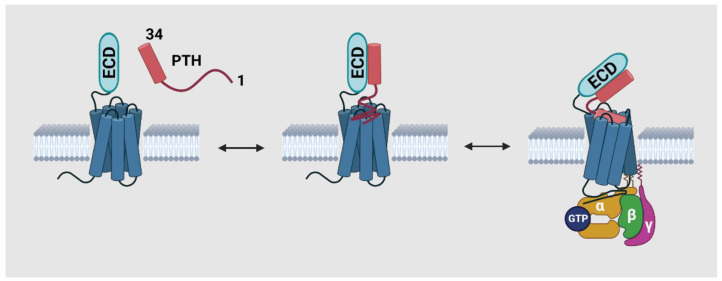
The two-site model describes the interaction between PTHrP/PTH and PTH1R interaction mechanism. In this model, PTHrP (or PTH) binds to the extracellular domain (ECD) of the PTH1R at the juxtamembrane region, and then interacts with the transmembrane domain of the receptor to induce intracellular signaling. This model is supported by experimental evidence showing that mutations in the juxtamembrane region of PTH1R impair the binding of PTHrP to the receptor and downstream signaling [52].

**Figure 4 cancers-15-01982-f004:**
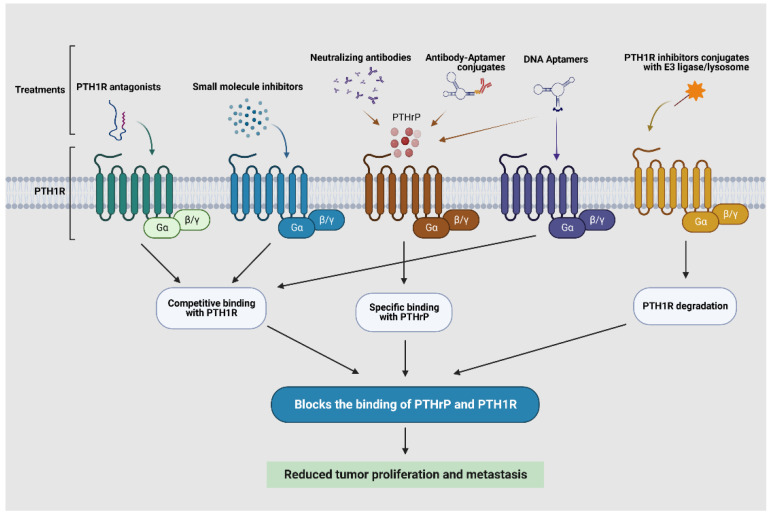
Current and prospective approaches to target PTHrP/PTH1R.

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
