# Peer review of "Parathyroid Hormone-Related Protein/Parathyroid Hormone Receptor 1 Signaling in Cancer and Metastasis"

_cancers, 2023, doi:10.3390/cancers15071982_

Round 1

Reviewer 1 Report

This review focuses on the cell signal initiated by the complex of parathyroid hormone related protein and parathyroid hormone receptor 1, in relation to cancer. It covers the background, current situation of investigation, and future research in this field. It is comprehensive and informative, and I recommend the publication of this manuscript in "cancers". However, I suggest several minor issues as listed below.

1. Title: Most journals recommend to avoid non-standard abbreviations in title.

2. Line 151: "ECD domain" ---> "ECD"

3. Figure 3: In right figure, GTP seems to be a part of polypeptide. GTP should be shown in different color.

4. Line 172: "Importin" ---> "importin"

5. Line 194: In "GaS-mediated", "S" should be in subscript.

6. Line 211: This paragraph explaining R0 and RG should be placed at the beginning of Section 2.3.

7. Figure 4: I understood that RG conformation is G-protein-binding conformation. If so, PRH1R in right panel should be in complex with Gs.

8. Line 294: "breast" ---> "Breast"

9. Line 299: "[92]96]" ---> "[92][96]"

10. Line 698: Ref. 114: Journal name etc. are missing.

Reviewer 2 Report

The authors reviewed existing data on the role of PTHrP-PTH1R axis in cancer and metastasis. Published data are controversial, so summarizing them has merit. Discussing possible future directions for therapeutic targeting of these molecules is also timely. Several factual errors and presentation issues should be corrected.

1.     Fig. 1. Explain what the green region is.

2.     Lines 124-126. Recent structural work showed that the PTH1R core (the cavity between transmembrane helices that opens upon receptor activation) engages both G proteins and arrestins. This is not solely the function of the cytoplasmic C-terminal element of PTH1R.

3.     Lines 133-135, Fig. 2, and the text below. G protein independence of PTH1R signaling to ERK via b-arrestins has not been shown. For a number of other GPCRs it was shown that b-arrestins cannot facilitate ERK activation in the absence of functional G proteins (Nat Commun. 2018 Jan 23;9(1):341).

4.     Lines 167-169. How can PTH1R by activated by nuclear PTHrP? Nuclear space through nuclear pores connects to the cytoplasm. PTH1R elements that bind PTHrP are extracellular or upon internalization intravesicular, i.e., they never are exposed on that side of the membrane. Being an integral membrane protein, PTH1R cannot be in solution, it always resides in some kind of a membrane. Also, how do the authors envision the presence of PTH1R in the nucleus, which is devoid of internal membranes. Explain.

5.     Lines 199-200. In view of what we know, the sentence “Upon ligand binding, the PTHrP/PTH1R complex is internalized and recruits β-arrestin to further promote cAMP signaling by activating ERK1/2” makes no sense. Neither does the next sentence.

6.     Lines 211-219. Either PTH1R is dramatically different from hundreds of other GPCRs, or this paragraph, as well as Fig.4, is incorrect. Explain.

7.     Lines 375-376. The structure of the complex of PTH1R with b-arrestin1 was recently deduced from cross-linking two proteins bearing unnatural amino acids in the environment of the living cell (Nat Commun. 2023 Mar 1;14(1):1151).

8.     Lines 379-380. D-amino acids change the geometry of the peptide. Stability can be increased by using N-methylated amino acids.

9.     Lines 381-382 and Fig. 5. To recruit E3 ubiquitin ligases, the molecule must be intracellular, as E3 ligases are only present inside the cell. Thus, the issue of cell permeability arises.

10.  Some editing is needed: line 63, “intermit” should be “intermittent”, as in line 74; line 69, “consist of” should be “comprise”; line 91, in “amino N-terminal, mid-region, and carboxy C-terminal domains” should be either amino-terminal, or N-terminal, as well as either carboxy-terminal or C-terminal, as below; line 99 and below, “NTS” should be “NLS”, as in legend to Fig. 1; line 185, “recycle” should be “recycling”; line 280, rephrase “cancer-educated”; etc.
